# Clinical Evaluation of a Fully-Automated High-Throughput Multiplex Screening-Assay to Detect and Differentiate the SARS-CoV-2 B.1.1.529 (Omicron) and B.1.617.2 (Delta) Lineage Variants

**DOI:** 10.3390/v14030608

**Published:** 2022-03-15

**Authors:** Dominik Nörz, Moritz Grunwald, Hui Ting Tang, Celine Weinschenk, Thomas Günther, Alexis Robitaille, Katja Giersch, Nicole Fischer, Adam Grundhoff, Martin Aepfelbacher, Susanne Pfefferle, Marc Lütgehetmann

**Affiliations:** 1Institute of Medical Microbiology, Virology and Hygiene, University Medical Center Hamburg-Eppendorf (UKE), 20246 Hamburg, Germany; d.noerz@uke.de (D.N.); grunwald.moritz@web.de (M.G.); hui-ting.tang@stud.uke.uni-hamburg.de (H.T.T.); celine.weinschenk@stud.uke.uni-hamburg.de (C.W.); kgiersch@uke.de (K.G.); nfischer@uke.de (N.F.); m.aepfelbacher@uke.de (M.A.); s.pfefferle@uke.de (S.P.); 2Leibniz Institute for Experimental Virology (HPI), Virus Genomics, 20251 Hamburg, Germany; thomas.guenther@leibniz-hpi.de (T.G.); alexis.robitaille@leibniz-hpi.de (A.R.); adam.grundhoff@leibniz-hpi.de (A.G.)

**Keywords:** SARS-CoV-2, Omicron variant, RT-qPCR, variant screening

## Abstract

Background: The recently emerged SARS-CoV-2 B.1.1.529 lineage and its sublineages (Omicron variant) pose a new challenge to healthcare systems worldwide due to its ability to efficiently spread in immunized populations and its resistance to currently available monoclonal antibody therapies. RT-PCR-based variant tests can be used to screen large sample-sets rapidly and accurately for relevant variants of concern (VOC). The aim of this study was to establish and validate a multiplex assay on the cobas 6800/8800 systems to allow discrimination between the two currently circulating VOCs, Omicron and Delta, in clinical samples. Methods: Primers and probes were evaluated for multiplex compatibility. Analytic performance was assessed using cell culture supernatant of an Omicron variant isolate and a clinical Delta variant sample, normalized to WHO-Standard. Clinical performance of the multiplex assay was benchmarked against NGS results. Results: In silico testing of all oligos showed no interactions with a high risk of primer-dimer formation or amplification of human DNA/RNA. Over 99.9% of all currently available Omicron variant sequences are a perfect match for at least one of the three Omicron targets included in the multiplex. Analytic sensitivity was determined as 19.0 IU/mL (CI95%: 12.9–132.2 IU/mL) for the A67V + del-HV69-70 target, 193.9 IU/mL (CI95%: 144.7–334.7 IU/mL) for the E484A target, 35.5 IU/mL (CI95%: 23.3–158.0 IU/mL) for the N679K + P681H target and 105.0 IU/mL (CI95%: 80.7–129.3 IU/mL) for the P681R target. All sequence variances were correctly detected in the clinical sample set (225/225 Targets). Conclusion: RT-PCR-based variant screening compared to whole genome sequencing is both rapid and reliable in detecting relevant sequence variations in SARS-CoV-2 positive samples to exclude or verify relevant VOCs. This allows short-term decision-making, e.g., for patient treatment or public health measures.

## 1. Introduction

The SARS-CoV-2 B.1.1.529 lineage (including the BA.1, BA.2 and BA.3 sublines) was first identified in November 2021 through whole genome sequencing from clinical samples in Botswana and classified as variant of concern (VOC) “Omicron” by the World Health Organization (WHO) shortly thereafter [1]. The Omicron variant features an unusually large number of mutations compared to previously prevalent lineages, over 30 of which are located in the Spike-gene (S-gene) and significantly reduce the efficacy of neutralizing antibodies generated through past infection or vaccination [2,3]. The Omicron variant drove a steep new wave of infections within the South-African region, the United Kingdom, and Denmark and has since become dominant worldwide, likely due to its ability to efficiently infect populations with a high degree of pre-existing immunity to previously prevalent SARS-CoV-2 variants such as Beta and Delta [4]. More recently, the BA.2 and BA.1.1 sublines see continued expansion within Europe, the former of which lacks certain sequence variances in the Spike-Gene (notably del-HV69-70) and the latter featuring an additional one (R346K), both of which have been shown to affect antibody efficacy [5].

Rapid PCR typing assays are warranted when encountering new variants, specifically regarding time and cost compared to whole-genome sequencing, to provide fundamentals for quick decision-making in the clinic and public health policy. It has been noted early on, that Omicron variant samples will present with “S-gene target failure” on the Thermofisher TaqPath SARS-CoV-2 assay [6,7,8] due to the HV69-70 deletion (except BA.2), similar to Alpha (B.1.1.7) and other lineages. Apart from multiple deletions in the N-terminal domain [9], the Omicron variant offers a wide range of S-gene single nucleotide polymorphisms (SNP) in functionally relevant regions such as the receptor binding domain [10] and furin cleavage site [11], which are well known from previous variants. Such SNPs can be detected by RT-PCR through different methods in order to predict lineages based on sequencing and epidemiological data [12,13,14].

The aim of this study was to compile a multiplexed RT-PCR assay for detection of four different Spike-gene mutations in order to differentiate Omicron and Delta variant samples on a fully automated high-throughput platform. This can serve to provide clinicians and public health officials with timely information about the presence or absence of relevant variants in individual patients.

## 2. Material and Methods

### 2.1. Assay Design

The Omicron variant features a number of mutations (SNPs and deletions) which have previously been found in other VOCs, e.g., del-HV69-70 and P681H (www.outbreak.info, accessed 24 December 2021). However, many of these are now accompanied by additional SNPs within potential probe regions, such as A67V or N679K. The inclusion of these additional variances allows for assays to be highly specific for the Omicron variant.

A set of previously described TaqMan-assays of our group [15,16] was modified for the respective target regions of the Omicron variant S-gene [16]. Briefly, assays were designed using PrimerQuest software (IDT) with probes being 12–20 bp in length, containing a triplet of locked nucleic acid (LNA)-bases at the SNP location and melting temperature being adjusted by including additional LNA-bases. If additional SNPs were present in the probe target region, these were covered with LNA-bases to improve discrimination. For the HV69-70 deletion, the probe sequence was modified to omit the affected bases and an LNA-base positioned at the A67V SNP (probe 1, “SDEL2”). For the furin-cleavage-site, probe-4 (“P681H”) also covers N679K with an LNA-base. By using this approach, we generated two Omicron “specific” targets (A67V + DEL69/70; and P681H + N679K). For sequence variants within the probe regions that are not covered by specific probes, blocker oligos were included in the assay [16].

In general, LNA bases allow for shorter Taqman-Probes by increasing melting temperatures, which is an established method for increasing sequence specificity [15,17]. 2′O-methyl RNA bases are placed at the penultimate position of every primer to reduce the formation of primer dimers. Blocker-oligos serve to depress off-target activity of specific probes, especially when discriminating for a single mismatch, thus allowing for lower RFI (relative fluorescence increase) thresholds.

The multiplex assay amplifies three regions of the SARS-CoV-2 S-gene: 102bp within the N-terminal domain (NTD) (probe 1: “SDEL2”), 353/80bp within the receptor-binding-domain (RBD) (probe 3, “E484A”), and 95bp at the furin-cleavage-site (probe 2: “P681R”, probe 4: “P681H”) (Figure 1). Primer sequences were modified with ambiguous bases to account for SNPs present in the B.1.1.529 lineage. For an overview about the presence or absence of relevant mutations in different VOCs, see Appendix A or, e.g., the outbreak.info website (www.outbreak.info, accessed on 26 February 2022).

**Figure 1 viruses-14-00608-f001:**
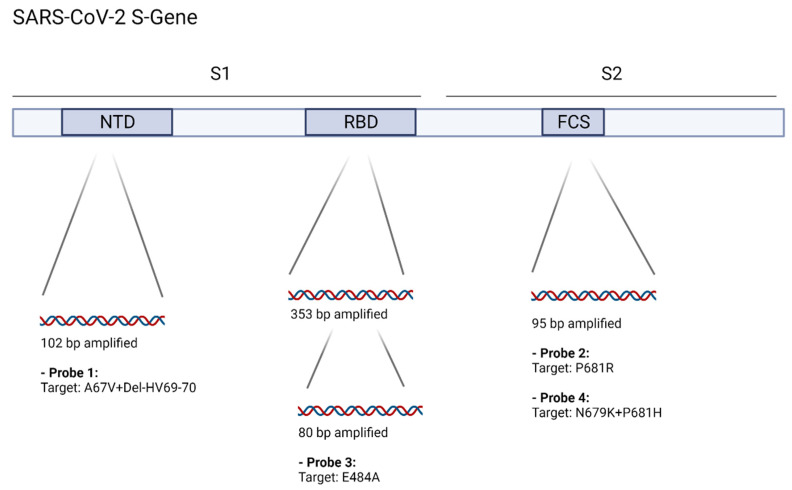
Target regions were chosen based on initial available Omicron variant sequences. The included primer sets are either still a 100% match or modified with ambiguous bases to amplify three target regions within the SARS-CoV-2 S-gene, containing deletions or SNPs in the Omicron variant. Four different LNA-Probes are used to specifically detect one or multiple sequence variances in order to allow classification. Delta variant sequences are expected to contain the P681R SNP (probe 2, “P681R”) but not the other tested variances. Omicron variant (Non-BA.2) sequences are expected to contain A67V + del-HV69-70 (probe 1, “SDEL2”), E484A (probe 3, “E484A”) and N679K + P681H (probe 4, “P681H”). The BA.2 lineage is expected to return negative for probe 1 but remain positive for the two other targets. NTD: N-terminal domain, RBD: receptor binding domain, FCS: Furin cleavage site. Image was created using Biorender software. All included oligos are listed in Table 1.

### 2.2. Inclusivity and Compatibility

Inclusivity of the primer/probe sets was determined by Roche diagnostics (Pleasanton, CA, USA) as part of a Utility channel support request. Sequences were excluded from the analysis, if the target regions could not be analyzed due to poor sequence quality. All Omicron sequences available (GISAID) until 20 December 2021 were included (13,462 for the NTD assay, 3472 for the RBD, and 11,910 for the furin cleavage site). All three target regions could be analyzed in 2984 sequences. The results are included in Appendix A. Notably, for those sequences in which all three target regions could be analyzed (n = 2984), 99.93% contained a perfect match for at least one of the three Omicron variant targets.

Performance of multiplex RT-PCR assays is jeopardized by strong oligo–oligo interactions, unspecific amplification of primer-dimers and amplification of non-target sequences, e.g., from human DNA/RNA, among other factors. For oligo–oligo interactions and risk of primer-dimer formation, all oligo sequences were analyzed against each other using OligoAnalyzer software (IDT, Coralville, IA, USA) (Appendix A). Five interactions were analyzed further due to high binding energy (RBD-484-fwd: RBD-484-fwd (2×), P681-fwd: P681fwd, RBD-484-fwd: P681R-probe, RBD-484-fwd: P681H-probe) (Appendix A). Of these, none show overlap at the 3′ ends of the respective oligos.

Matches in human DNA/RNA with a risk to lead to efficient amplification were analyzed using Primerblast (NCBI, USA). Two human RNA targets, 672bp and 969bp in length can potentially be amplified by included primers due to a perfect match of 10 or more bases at the 3′-ends (Appendix A).

### 2.3. Utility Channel Test Setup

Primers and Probes were added to MMX-R2 reagent and loaded into cobas omni Utility Channel cassettes (Roche diagnostics, Rotkreuz, Switzerland) according to instructions by the manufacturer. The test is referred to as SCOV2_OMIC_VOC-UCT. The complete run protocol is shown in Table 2.

All cobas omni Utility Channel tests contain a spike-in full-process-control (internal control, IC) by default. Primers and probes are preloaded in MMX-R2 reagent, and the spike-in RNA target is added automatically during extraction. The internal control is detected in Channel 5 for each reaction and is functionally identical to commercial CE-IVD tests by Roche Diagnostics on the cobas 6800/8800 instruments.

Loaded cassettes are placed into the system for processing. No further manual steps are needed to perform a test.

### 2.4. Analytic Sensitivity, Inter-, and Intra-Run Variance

For evaluation of analytic performance, cell culture supernatant of a clinical isolate of SARS-CoV-2 Omicron variant was used to create dilution series. Briefly, 500 µL medium (UTM) of a confirmed SARS-CoV-2 (Omicron variant) positive nasopharyngeal swab sample (UTM, Mantacc, Shenzhen, China) were used for virus isolation on Vero E6 cells (ATCC^®^ CRL-1008) as described previously [18].

Lineage of the isolate was confirmed by NGS before and after virus isolation (GeneBank OL960487). SARS-CoV-2 WHO standard (NIBSC, Potters Bar, UK) was used in conjunction with the cobas SARS-CoV-2 CE-IVD test (Roche Diagnostics, Rotkreuz, Switzerland) to create a quantified reference stock (in IU/mL). For the P681R target, a clinical Delta-variant sample (confirmed by NGS) was normalized to WHO standard using the same method. Analytic limit of detection (LoD) was determined by 2-fold dilution series (8 steps, 8 repeats, performed on a Hamilton STARlet IVD liquid handler), run on a cobas6800 instrument with the SCOV2_OMIC_VOC-UCT. Then, 95% probability of detection and confidence intervals (CI95%) (probit analysis) were determined with MedCalc Software (Ostend, Belgium).

For inter- and intra-run variability, a set of 10-fold dilutions (4 steps, 5 repeats) were run with the test on three separate days.

For exclusivity testing, a panel of 27 external controls or clinical samples was tested, containing various respiratory viruses, notably endemic human coronaviruses, Middle East respiratory syndrome-related Coronavirus (MERS-CoV) and SARS-CoV (2003, Frankfurt-1, AY291315).

### 2.5. Clinical Performance

A total of 244 predetermined clinical remnant samples (by qPCR and NGS if positive for SARS-CoV-2 RNA) were run with the SCOV2_OMIC_VOC-UCT.

Of these samples, 51 were predetermined negative for SARS-CoV-2 RNA by the cobas SARS-CoV-2 CE-IVD test. A total of 75 samples were positive for non-Delta, non-Omicron SARS-CoV-2 lineages (notably 50 B.1.1.7 samples); 54 samples were predetermined Delta variant (B.1.617.2-like or AY.4-like lineages), and 64 samples were predetermined as Omicron variant (B.1.1.529 lineage, 42 BA.1-like and 22 BA.2-like). All lineages were assigned based on whole genome sequencing, carried out at the Leibniz Institute for Experimental Virology (HPI, Hamburg), as part of the Hamburg Genome Surveillance Project (https://www.hpi-hamburg.de/en/, accessed on 14 March 2022).

## 3. Results

### 3.1. Analytic Performance

Analytic LoD was determined by 2-fold dilution series (8 steps, 8 repeats) as 19.0 IU/mL (CI95%: 12.9–132.2 IU/mL) for target 1: A67V + del-HV69-70; 105.0 IU/mL (CI95%: 80.7–129.3 IU/mL) for target 2: P681R; 193.9 IU/mL (CI95%: 144.7–334.7 IU/mL) for target 3: E484A; and 35.5 IU/mL (CI95%: 23.3–158.0 IU/mL) for target 4: N679K +P681H (See Table 3). Probit plots are included in Appendix A.

The multiplex test returned negative for all clinical and external control samples of the exclusivity set, notably, SARS-CoV, MERS-CoV, and endemic human coronaviruses (see Appendix A).

### 3.2. Clinical Performance

A total of 244 qPCR and NGS predetermined (if positive) clinical remnant samples were run with the test to assess clinical performance. A total of 193 clinical samples were positive for SARS-CoV-2 RNA. All four targets (225 sequence variances in total, SNPs, and deletions) were correctly identified by the multiplex assay (Table 4). The clinical sample set notably included SARS-CoV-2 lineages with P681H (non-N679K), Del-HV69-70 (non-A67V), E484K and E484Q sequence variances (e.g., B.1.1.7 (Alpha), B.1.351 (Beta), and B.1.617.1 (Kappa)). A complete list can be found in Table 5. Exemplary amplification curves are displayed in Appendix A.

## 4. Discussion and Conclusions

In this study, we presented a multiplexed screening assay with three independent targets for the Omicron variant (first: A67V + del-HV69-70, second: E484A, and third: N679K + P681H) and one for the delta variant P681R, thus allowing easy discrimination between these two prevalent lineages. 

The S-gene target dropout (due to the HV69-70 deletion) observed on the widely used TaqPath SARS-CoV-2 assay (Thermofisher) has reportedly been used for tracking the expansion of the B.1.1.7 lineage [6] and is now used again for the same purpose with the novel Omicron (Non-BA.2) variant [8]. However, due to the unspecific nature of assay dropouts and the prevalence of NTD deletions in non-Omicron variants, it is preferable to specifically detect SNPs or combinations of SNPs that are conserved within the lineage in question [16]. There now exists a wide range of lab-developed and commercial solutions for this purpose [12,13,14], although their suitability for application on a large scale varies as many available protocols are highly manual and require careful interpretation of every individual result. Furthermore, the Omicron variant is reported to feature a lot of heterogeneity within well-established target regions; this, however, seems to be largely a result of low sequence quality or faulty annotations as NGS struggles with large numbers of mismatches for amplicon-primers, especially within the RBD region.

For the multiplex-test described in this study, over 98% of available Omicron sequences have perfect sequence identity with the individual SNP/deletion-assays, when only including those that have valid sequences for the respective regions. Only 0.07% of currently available Omicron sequences would return negative on all three assays.

Of note, the BA.2 subline (sometimes dubbed the “stealth variant”) lacks the A67V and del-HV69-70 mutations, thus leading to a negative result for the NTD target, while retaining positivity for E484A and N679K + P681H; a pattern that can be utilized to distinguish contemporary BA.2 from other Omicron sublines.

Still, the evolution of the Omicron variant remains in an early stage, and distinct sublines may emerge, or individual SNPs may be lost within the coming months. It is therefore particularly important to continuously monitor emerging sequences for mismatches in the oligo-set. In any case, definitive assignment of lineages can only be carried out based on whole genome sequencing results; PCR based typing can and should be used as a complement but not replace NGS. Furthermore, it should be noted that the assay is not intended or suited as a first-line SARS-CoV-2 assay but as a typing-test following detection of SARS-CoV-2 RNA by established methods. Results of individual targets should be considered valid only if the clinical sample tested contains SARS-CoV-2 RNA at concentrations above the respective LoDs.

As such, the multiplex assay we present here can be an important asset allowing clinicians and public health officials to rapidly act on suspected cases or outbreaks. Rapid differentiation of newly diagnosed infections may become particularly relevant for this newly emerged lineage due to potential implications for treatment or quarantine mandates. For instance, the often life-saving monoclonal antibody preparations seem to have largely lost their effectiveness against the Omicron variant in cell culture neutralization experiments [2,3,19]. Further studies have demonstrated how pseudovirus neutralization of frequently used monoclonal antibodies such as Regencov is greatly diminished for the Omicron variant-Spike protein, while, e.g., Sotrovimab remains little affected for BA.1-Spike but substantially loses efficacy against BA.2-Spike [5]. Rapid variant screening may be of value for allocating monoclonal antibody treatments in the future. From a public health perspective, local Omicron variant outbreaks may pose a more urgent need for decisive intervention, due to Omicron’s ability to efficiently spread in pre-immunized populations [4]. It can further help classifying low viral-RNA load samples, for which NGS often fails to obtain viable sequences [20]. As described above, the A67V + del-HV69-70 and N679K + P681H components of the multiplex assay feature analytic sensitivities similar to currently available diagnostic SARS-CoV-2 tests (e.g., between 30–50 dcp/mL for the cobas SARS-CoV-2 test [21] and can detect relevant sequence variances even in samples with very low viral-RNA loads.

The adaptation of variant screening assays for automated high-throughput platforms further enables laboratories to efficiently predict SARS-CoV-2 lineages for large numbers of samples and in a timely manner, thus serving as an important complement to whole-genome sequencing based surveillance programs.

## Figures and Tables

**Table 1 viruses-14-00608-t001:** Primer-, probe-, and blocker sequences of the multiplex assay are listed. Oligos were custom made by Ella Biotech (Fürstenfelbruck, Germany). Indicated final concentrations refer to the final oligo concentrations within the reaction mix. 2′O-methyl-RNA bases are indicated as “OMe-X”. LNA bases are indicated as “+X”).

Oligo Type	Oligo Name	Sequence 5′–3′	Final Concentration [nM]
Primers	NTD fwd	TCA ACT CAG GAC TTG TTC T(OMe-U)A C	400
	NTD rev	TGG TAG GAC AGG GTT AT(OMe-C) AAA C	400
	RBD-452 fwd	GAT T(+C)T AAG GTT GGT GG(2OMe-U) AAT	400
	RBD-484 fwd	CTA TCA GGC CGG TAR (2OMe-C)A	400
	RBD-484-rev	GTC GGA AAC TAT ATG ATC GTA AA(OMe-G) G	400
	RBD-univ-rev	AGT TGC TGG TGC ATG TA(OMe-G) AA	400
	FCS fwd	TGC AGG TAT ATG CGC TAG T(OMe-U)A	400
	FCS rev	GTG ACA TAG TGT AGG CAA TGA (OMe-U)G	400
Probes	A67V-del69-70 probe	Atto425- TGG TCC CAG A(+G)A T(+A)(+A) C(+A)T -BHQ1	50
	E484A probe	YakYellow- AT(+G) GTG TT(+G) (+C)(+A)G (+G)TT -BHQ1	50
	P681R probe	FAM- A(+T)T CT(+C) (+G)(+T)C GGC G -BHQ1	50
	N679K-P681H probe	Atto620- A(+G)T CT(+C) (+A)(+T)C GG(+C) G -BHQ2	50
Blockers	E484WT blocker	AT(+G) GTG T(+T)(+G) (+A)AG (+G)TT -C3-Spacer	50
	E484K blocker	AT(+G) GTG T(+T)(+A) (+A)AG (+G)TT -C3-Spacer	50
	E484Q blocker	AT(+G) GTG T(+T)(+C) (+A)AG (+G)TT -C3-Spacer	50
	P681WT blocker	TAA (+T)TC T(+C)(+C) (+T)CG GCG -C3-Spacer	50

The “NTD” primer set and “A67V-del69-70”-probe are derived from a published assay by Zhen et al. [7]. Oligos used in this study were custom made by Ella Biotech (Fürstenfeldbruck, Germany).

**Table 2 viruses-14-00608-t002:** Cobas omni Utility Channel run protocol for the SCOV2_OMIC_VOC-UCT. RFI (relative fluorescence increase) cut-offs are used to achieve specificity for a 100% sequence match for respective Taqman-probes.

**Software Settings**					
**Sample Type**	**Swab (400 µL)**				
**Channels**	**1: SDEL2**	**2: P681R**	**3: E484A**	**4: P681H**	**5: IC**
RFI	1.8	2.5	2.8	2.5	2
**PCR cycling conditions**					
	**UNG incubation**	**Pre-PCR step**	**1st measurement**	**2nd measurement**	**Cooling**
No. of cycles	Predefined	1	5	45	Predefined
No. of steps		3	2	2	
Temperature		55 °C; 60 °C; 65 °C	95 °C; 55 °C	91 °C; 58 °C	
Hold time		120 s; 360 s; 240 s	5 s; 30 s	5 s; 25 s	
Data acquisition		None	End of each cycle	End of each cycle	

**Table 3 viruses-14-00608-t003:** LoDs were determined by serial dilution of a quantified Omicron variant cell culture stock and a clinical Delta variant sample, using the SARS-CoV-2 WHO standard (NBSCI, UK) as reference. Dilution series were generated automatically using a Hamilton STARlet IVD liquid handler. A 95% probability of detection was calculated using medcalc software. * P681R LoD constitutes a separate experiment using a quantified clinical sample of the B.1.617.2 lineage.

SARS-CoV-2 Omicron (B.1.1.529) and Delta (B.1.617.2) Variants					
Step	IU/ml	SDEL2: Pos/Rep	E484A: Pos/Rep	P681H: Pos/Rep	P681R: Pos/Rep *
1	500.00	8/8	8/8	8/8	8/8
2	250.00	8/8	8/8	8/8	8/8
3	125.00	8/8	4/8	8/8	7/8
4	62.50	8/8	4/8	7/8	8/8
5	31.25	8/8	1/8	7/8	4/8
6	15.63	7/8	0/8	6/8	5/8
7	7.81	5/8	0/8	5/8	0/8
8	3.91	3/8	0/8	2/8	0/8

**Table 4 viruses-14-00608-t004:** SCOV2_OMIC_VOC-UCT results for the clinical sample set. Each target/channel is analyzed individually.

Target	Result	SNP Positive	SNP Negative	Agreement
A67V + del-HV69-70	Positive	42	0	100%
	Negative	0	202	100%
P681R	Positive	55	0	100%
	Negative	0	189	100%
E484A	Positive	64	0	100%
	Negative	0	180	100%
N679K + P681H	Positive	64	0	100%
	Negative	0	180	100%

**Table 5 viruses-14-00608-t005:** Lineages included in the clinical sample set, as determined by whole genome sequencing.

Clinical Sample Set—Included Lineages		
SNP Set	Lineage	Number
All negative	B.1.1.7-like (Alpha)	50
	B.1.177	10
	B.1.221	6
	B.1.1.29	6
	C.36.3	1
	B.1.351 (Beta)	1
P681R	B.1.617.2-like	45
	AY.4-like	9
	B.1.617.1 (Kappa)	1
A67V, del-HV69-70 E484A N679K, P681H	B.1.1.529 (Omicron) BA.1-like	42
E484A N679K, P681H	B.1.1.529 (Omicron) BA.2-like	22

## Data Availability

In cases where sequencing was performed, all sequences were made publicly available at GISAID. All data is available upon request.

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
