# Peer review of "Clinical Evaluation of a Fully-Automated High-Throughput Multiplex Screening-Assay to Detect and Differentiate the SARS-CoV-2 B.1.1.529 (Omicron) and B.1.617.2 (Delta) Lineage Variants"

_viruses, 2022, doi:10.3390/v14030608_

Round 1

Reviewer 1 Report

Nörz et al. describe a new multiplex to differentiate the SARS-CoV-2 B.1.1.529 (Omicron) and B.1.617.2 (Delta) lineages. The manuscript is well written and organized, the data is relevant for other laboratories. The analyses are well performed and the results well discussed. I have no major issues to address.

Reviewer 2 Report

The study from Nörz et al. presents the results of a multiplex variant screening RT-PCR assay developed and validated on cobas 6800/8800. By targeting 3 different regions in the SARS-CoV-2 S gene and by using highly specific probes, this assay allows the discrimination between Delta and Omicron variant and correlate very well with whole genome sequencing. Although many other tests are now available and widely used to detect SARS-CoV-2 Delta and Omicron variant, this assay is interesting due to its use on automated high-throughput platform. Unfortunately, in the ever-changing epidemiological scenario of SARS-CoV-2, the Delta variant has been almost completely replaced by the Omicron and it is hardly circulating in Europe anymore. This limit the clinical interest of such assay now.  On the other hand, this assay could be useful to monitor the spread of BA.1 and BA.2 Omicron and to detect the emergence of possible new variants.

After reading the manuscript I have some comments:

Major points:

  • In the current context with the disappearance of the Delta variant and the progression of the BA.2 variant, it seems essential to me to validate this assay with clinical samples of Omicron BA.2 variants.
  • While this assay was developed to differentiate between the Delta and Omicron variants, the whole performance validation part is performed only for the detection of the Omicron variant. The determination of the LoD of the P681R target using cell culture supernatant of a Delta variant is missing and should be added.
  • A limitation of this assay is that it does not target a conserved region like diagnostics assays. This would allow to use this assay as a first line RT-PCR assay. Moreover, with this assay we cannot distinguish new emerging SARS-CoV-2 variant from Delta or Omicron variants with low viral loads that would not be detected by the assay. Both would lead to negative results. Some laboratory use TMA or LAMP assays as first line diagnostic assay that do not provide Ct information regarding viral load that could help distinguish new variants from low positive patients.

Minor points:

  • Keywords are missing
  • The introduction should be actualized, the Delta variant is not dominant any more. (L47) Reference must be updated too. A lot of not peer reviewed article are cited whereas they coud be replaced by accepted peer reviewed articles. For example ref 2 : Planas et al; Nature 2021 (DOI: 10.1038/s41586-021-04389-z.). A reference on the association of thermofisher S gene target failure profile and Omicron is missing. Ref 13 and 14 are the same.
  • The design of the assay is quite complex and difficult to follow. I think that a figure describing the mutations of the main VOCS (Alpha, Beta, Gamma, Delta, Omicron BA.1 and BA.2) on the targeted regions by the assay would help the readers. Moreover in the material and methods the utility of primers with OMe RNA bases, probes with LNA bases, and blockers should be further detailed.
  • Supplementary figure 3 should be updated with the amplification curves on the 4 channel for (1) Omicron BA.1; (2) Omicron BA.2; (3) Delta; (4) Other SARS-CoV-2 lineages

Reviewer 3 Report

In this manuscript, Dominik Nörz et al. developed a novel assay system to distinguish SARS-CoV-2 Omicron and Delta variants. The topic of the manuscript is timely and relevant for the contemporary COVID-19 pandemic. This reviewer does not have any substantial amendments to suggest.

Reviewer 4 Report

The manuscript by Nörz et al., was a relatively comprehensive research article on the potential roles of the genetic variants for detection of COVID-19 viruses. The authors focused on the SARS-CoV-2 Omicron and tested using primer-based PCR methods. Their bioinformatics analysis further verified the existence of viral sequence in clinical samples. Investigations were well designed and data was well collected. There were some moderate concerns:

  • Rationale of the test to “differentiate Omicron and Delta variant” (such as in Page 2 Lines 59-61) was unclear.
  • Based on the Table 3 and related experiments, the detection limit/calculation was unknown.
  • Indicate whether co-infection may exist in the samples and how to confirm in the authors’ research.

Round 2

Reviewer 2 Report

I would like to thank the authors who provided clear answers to my questions and modified the manuscript accordingly.
The manuscript has been sufficiently improved to warrant publication in Viruses.